# Nanocomposite Materials Based on Polylactide and Gold Complex Compounds for Absorbed Dose Diagnostics in BNCT

**DOI:** 10.3390/ijms242216492

**Published:** 2023-11-18

**Authors:** Vladislav Potseleev, Sergey Uspenskii, Elena Trofimchuk, Anastasia Bolshakova, Anna Kasatova, Dmitrii Kasatov, Sergey Taskaev

**Affiliations:** 1Enikolopov Institute of Synthetic Polymeric Materials, Russian Academy of Sciences, 70, Profsoyuznaya Str., 117393 Moscow, Russia; vladislav.potseleev@chemistry.msu.ru; 2Faculty of Chemistry, Moscow State University, 1, Leninskie Gory, 119991 Moscow, Russia; elena_trofimchuk@mail.ru (E.T.); nastya@polly.phys.msu.ru (A.B.); 3Budker Institute of Nuclear Physics, Siberian Branch of Russian Academy of Sciences, 11 Lavrentieva, 630090 Novosibirsk, Russia; yarullinaai@yahoo.com (A.K.); kasatovd@gmail.com (D.K.)

**Keywords:** nanocomposite, polylactide, cancer therapy, gold complex, drug delivery, release, biocompatibility, hydrolysis

## Abstract

In this study, approaches to the synthesis of complex compound of gold with cysteine [AuCys]_n_ for measuring absorbed dose in boron neutron capture therapy (BNCT) were developed. The dependence of the complex particle size on pH were established. Nanocomposite materials based on polylactide containing [AuCys]_n_ particles with an average size of about 20 nm were obtained using the crazing mechanism. The structure of obtained materials was studied by electron microscopy. The release kinetics of [AuCys]_n_ from polymer matrix were investigated. Release of [AuCys]_n_ from the volume of the polymeric matrix had a delayed start—this process began only after 24 h and was characterized by an effective rate constant of 1 μg/h from a 20 mg composite sample. At the same time, in vitro studies showed that the concentration of 6.25 μg/mL was reliably safe and did not reduce the survival of U251 and SW-620 cells.

## 1. Introduction

Boron neutron capture therapy (BNCT) is one of the promising areas in radiotherapy [1]. The main feature of BNCT is the possibility of a local increase in the absorbed dose in the tumor with virtually no effect on healthy tissues [2]. Compounds containing the boron-10 isotope are used in BNCT, and heavy elements, such as gold, can be used to measure the absorbed dose [3,4]. Predominantly, such compositions consist of gold nanoparticles (NPs) in combination with boron compounds [4]. However, gold NPs are not safe for the body [5]. The main challenges associated with using gold NPs in BNCT are their tendency to accumulate primarily in the liver and spleen, as well as the difficulty in later removing them from a biological organism [5]. The synthesis of drugs that are safe for the body, have high neutron absorption and the ability to accumulate in tumor cells and can be successfully removed from the body is the main scientific problem that needs to be solved.

Gold is safer for the organism if it is mainly in molecular form, for example, in the form of chelates. It is generally accepted that the biological activity of microelements and their wide engagement in all major metabolic reactions and in cellular chemistry depend on their chelating properties [6,7,8]. Reactions of chelate structure formation are the basis for the formation of reactive molecules, transformation of metabolites into structurally organized specific systems, formation of immunity and other immunodynamic and biodynamic processes of the organism [9,10]. Gold can be in the form of chelate compounds, such as complex or coordination compounds with sulfur-containing molecules. Gold(I) coordination complexes are most stable when sulfur-containing molecules (HSR and SR_1_R_2_) are used as ligands. Some of these compounds [Au(I)-SR]_n_ are used in medicine as drugs for rheumatoid arthritis and other diseases [11,12,13]. This demonstrates their relative safety for use as a medicine [14,15,16,17,18]. Gold complex compounds can be obtained by the interaction of gold salts (I) or (III) directly with HSR and SR_1_R_2_ according to the approach reported in [19]. The process of formation of complex compounds of gold with cysteine results in the formation of supramolecules of submicron size [20]. However, for medical applications, it is necessary that the size of the particles should be less than 100 nm. To solve this problem, almost the only option is to limit the particle size in the pores of a specific size. In this paper, we propose the use of a nanoporous polymer matrix, as such a limiter.

In the treatment of some cancer types, such as glibostoma and glioma, polymer composites such as Gliadel^®^ plate (MGI Pharma, Inc., Bloomington, IN, USA) are available. After removal of a malignant neoplasm, the patient is still at risk of recurrence, since it is quite difficult to resect all tumor cells. The plates are placed directly into the area of the removed tumor and dissolve for several weeks.

It is worth noting that the use of biodegradable implants will help to preserve mechanical strength, functionality and stability of the organ after tumor destruction and during tissue regeneration. The implant will gradually degrade in the process of organism recovery, and its decay products in the form of safe compounds will be naturally excreted from the organism. Thus, implantable drug delivery systems based on biodegradable matrices have been used in medical practice for a long time, and the field itself is actively developing.

In this work, a system for encapsulating and forming gold complex compounds within the pores and fibrils of the polymer is proposed for the first time to limit the size growth of the supramolecules. The creation of a polymer-based nanocomposite material will additionally allow gold compounds to be stored in nanoscale form to maintain their uptake by tumor cells in case of relapse. In this work, polylactide (PLA) was chosen as a polymer matrix, and the compound of gold with cysteine [AuCys]_n_ was chosen as a chelate compound. PLA is a polymer of lactic acid, which is extensively used in pharmaceutical and medical applications due to its biocompatibility and biodegradability [21,22].

Our study is the first to propose a strategy for encapsulation and formation of complex gold salt in the form of supramolecules of a given size in the pores of a biodegradable and bioresorbable polymer based on PLA to solve the dosimetry problem in the BNCT technique.

The initial step of this work is the construction of model experiments to evaluate the efficiency of the chosen encapsulation methods to quantify the release of encapsulated gold compounds and their cellular safety. The aim of this work is to develop approaches to encapsulate gold coordination compounds in a polylactide matrix, to evaluate the release of these gold compounds under different conditions over time and to determine the safety of using this composite. 

## 2. Results and Discussion

It is known that almost all [Au(I)-SR]_n_ substances (where -SR is an organic sulfur-containing ligand, for example, cysteine Cys) in an aqueous solution can be cyclic or linear chains [23], but the length (n) of such complexes is often insufficiently studied. The structure of myochrysin (sodium aurothiomalate), which is a cyclic tetramer according to [24,25], is the most studied. However, in an alkaline medium and in the presence of an excess of HSR, [Au(I)-SR]_n_ complexes can form soluble [Au(SR)_2_]^−^ compounds [26].

Since [AuCys]_n_ is still poorly studied and there is no clear description of how [AuCys]_n_ behaves in solution, it was necessary to study the condition of [AuCys]_n_ in aqueous solutions with different pH. During synthesis, [AuCys]_n_ forms a colloidal system that is stable from 3–10 min (in acidic and neutral media) to 14 days (in alkaline media). In this work, the dependence of the average particle size of [AuCys]_n_ formed by the interaction of aqueous solutions of HAuCl_4_ and *L*-cysteine on the pH of the solution was first determined (Figure 1). It can be seen that the particle size decreased as the pH of the medium increased with the addition of NaOH solution. The higher the concentration of OH- ions, the larger the -SR particles, which were able to form a more stable chelate complex [Au(SR)_2_]^−^ by interacting with gold atoms located on the surface of the nanoparticles of the [Au(I)-SR]_n_ complex and exchanging ligands. As a result, the average particle size of [Au(I)-SR]_n_ decreased to about 350 nm in pH 7 medium, and about 100 nm in pH 11.

It is worth noting that for use in BNCT, the drug must be in a neutral medium. Unfortunately, under these conditions, [AuCys]_n_ forms a colloidal solution, stable for only a few minutes, then [AuCys]_n_ turns into a precipitate. The size of the particles in such conditions is of the order of 350 nm, which complicates their use in cancer diagnosis and therapy, since the particle must enter the cell before irradiation. Theoretically, the particle size of gold salts is important and crucial in cellular uptake. Particles smaller than 100 nm are best absorbed by the mechanism of pinocytosis and endocytosis [26]. However, a compound with a particle size smaller than 100 nm has not been previously obtained in pH 7 medium [20,27,28]. Therefore, the next step in this work was to create an implantable film material containing the [AuCys]_n_ complex with particle sizes smaller than 100 nm, which could be used to deliver this compound to the tumor. PLA is known to be an excellent candidate for biomedical applications, particularly in the development of drug delivery systems. Therefore, the biocompatible polymer PLA was chosen for this purpose, and PLA-based nanocomposites containing [AuCys]_n_ were prepared.

There are many ways to obtain composites based on polymer and inorganic additives. The most appropriate way to produce nanocomposites based on PLA and gold complexes is the synthesis of [Au(I)-SR]_n_ in a previously prepared nanoporous film, since in this case the particle size of the complex is limited by the pore size, and the process itself can be carried out under room temperature conditions. A rather simple and effective way to obtain nanoporous films is the crazing mechanism [29]. Crazing occurs in the process of uniaxial stretching of polymer films in an adsorption-active medium (AAM), which reduces the excess surface energy of the polymer, resulting in the formation of a fibrillar porous structure. In this case, the pores are filled with the surrounding liquid medium, which may contain a functional filler or its precursor as dissolved substance. The characteristics of porous matrices can vary within wide ranges, depending on the nature of the medium in which crazing occurs, the drawing rate, and temperature.

To obtain stable open-porous matrices based on PLA, the initial polymer films were preliminarily crystallized at a temperature of 50 °C for 45 min. After heating under these conditions, the degree of PLA crystallinity determined by differential scanning calorimetry (DSC) was about 30–40%, and the average crystallite size determined from X-Ray Diffraction (XRD) data were 14–16 nm (as coherent scattering region). It was previously suggested [21] that uniaxial stretching of such partially crystalline PLA films in 95% ethanol proceeded by the mechanism of delocalized crazing. Figure 2 shows the dependence of the effective volume porosity of the PLA crystalline film versus the tensile strain. It can be seen that the porosity gradually increased, reaching its maximum value around 40 vol.%. The atomic force microscopy (AFM) study of the morphology of such porous PLA films with a tensile strain of 100–150% shows (Figure 3) that during stretching, a system of small crazes about 100 nm wide appeared, which were filled with oriented separated fibrils about 20 nm in diameter, between which nanosized pores were found. This image corresponds to the structure formed by the mechanism of delocalized crazing in partially crystalline polymers, such as polyethylene and polypropylene, previously described in some works [30,31].

The resulting porous PLA films were also studied by scanning electron microscopy (SEM) (Figure 4). It can be clearly seen that a fairly homogeneous structure at the micron level was formed throughout the entire volume of the polymer film during stretching. Furthermore, as it was previously shown [21], it had an open-porous structure, and it could be filled with filler throughout the entire volume.

The introduction of [AuCys]_n_ into PLA films was carried out by in situ synthesis of the complex in the pores of the matrix using the countercurrent diffusion methodology previously described in detail in the [32]. In this case, the chemical reaction and the formation of filler particles took place only in the pores of the matrix. As a result, the particles were immobilized and encapsulated in the polymer carrier. The materials obtained were white films containing 6.5 wt% of the filler [AuCys]_n_. SEM and transmission electron microscopy (TEM) micrographs of the composites are presented in Figure 5a,b,d. The formation of the [AuCys]_n_ crystalline phase in the PLA films was confirmed by the electron diffraction pattern of the nanocomposite (Figure 5c), which corresponded to that obtained for pure [AuCys]_n_. The interplanar distances for the crystalline phase of the complex determined from the standard gold sample were 0.37, 0.32, 0.27, 0.23, 0.2 and 0.17 nm.

It can be seen that [AuCys]_n_ was mainly concentrated on the surface (Figure 5a) and in the near-surface layer (Figure 5b) of the PLA matrix with a thickness of about 5 μm. At the same time, the filler comprised submicron-sized particles that formed larger agglomerates. Closer to the center of the composite volume, the concentration of filler decreased strongly, and it was found only as individual discrete spherical particles with sizes of 100–200 nm. It is important to note that these submicron [AuCys]_n_ particles also appeared to be aggregates consisting of nanoparticles of about 20 nm in size, as can be seen from the TEM micrograph of a thin section of the composite (Figure 5d). This fact seems to be a good result, since particles with sizes smaller than 100 nm are most suitable for BNCT applications. It is important to note that the encapsulated [AuCys]_n_ particles did not significantly affect the properties of the polymer matrix, such as its thermal properties. Figure 6 shows the DSC curves of the original PLA matrix and the matrix containing [AuCys]_n_.

It can be seen that these curves are similar. Their form is typical for partially crystalline polymers. They show one intense melting peak at a temperature of about 168.5 °C. The melting enthalpy ΔH_melt_ determined from the area of the melting peak was also similar for both samples, and was 32 J∙g^−1^ for the original sample and 35 J∙g^−1^ for the composite. From the ΔH_melt_ values, the degrees of crystallinity of the samples were determined, which were also similar and equal to 34% and 37%, respectively.

For materials containing functional additives, an important task is to study their release kinetics. Preliminarily characterized composite samples weighing 20 mg and containing 6.5 wt% [AuCys]_n_ were investigated for their ability to release the functional additive in a medium simulating the conditions of the human body. As such a model medium, water was chosen and maintained at pH 7 and at a temperature of 37 °C throughout the experiment as a simulation of the body medium. Additionally, the gold compound release experiment was carried out in sodium phosphate buffer solution with pH 9 at 37 °C, since the solubility of [AuCys]_n_ was significantly higher in alkaline medium. It was hypothesized that this could lead to a more intense release of the additive, allowing a more accurate determination of its release concentration.

It is known from literature data [33] that the [AuCys]_n_ compound absorbs in the UV region. Figure 7 demonstrates the characteristic absorption spectrum of [AuCys]_n_ in the UV range obtained during the study of the release kinetics. The values of optical density close to the maximum of the most intense peak (210 nm) were used to plot the calibration graph (Figure 7b).

According to the Beer–Lambert law for two substances in solution absorbing light of a given wavelength, the optical density is the sum of the optical densities of these substances $A=\varepsilon_{1}lC_{1}+\varepsilon_{2}lC_{2}$, where ε is the light absorption coefficient, l is the thickness of the light absorbing layer, and C is the concentration of optically active substance. Through the points of values of optical densities of standard solutions by means of the least squares method, the calibration graph was plotted, which is described by the linear function $A=\left(19500\pm 1100\right)C+\left(0.0568\pm 0.0011\right)$. Further, the kinetics of [AuCys]_n_ release from the obtained composites were studied using the obtained calibration graph, presented in Figure 7b.

Figure 8 and Figure 9 shows the time dependences of the optical density of the solution at pH 7 and pH 9, in which [AuCys]_n_ is released from PLA films. It can be seen that in the first hours there was a release of a small amount of functional additive. This was due to the transfer into solution of available [AuCys]_n_ particles, which were on the surface of the samples and were rather poorly bonded to it. Then the concentration of [AuCys]_n_ in solution remained almost constant for several hours. Due to the occurrence of hydrolysis, the surface of the PLA film appeared to become more friable. Thus, [AuCys]_n_, which was in the near-surface layer, began to be released into the medium. When the solubility limit of [AuCys]_n_ was reached, the concentration in the solution remained constant.

The results obtained for the kinetics of [AuCys]_n_ release over 37 days were approximated using the Gallagher and Corrigan model [34], the equations of which are presented in Table 1. The Gallagher and Corrigan model is a mathematical model describing the proportion of drug released from a biodegradable polymer system. The kinetic profile described by the Gallagher–Corrigan equation (Table 1) includes a starting “burst release” of drug unbound to the drug matrix and a subsequent slow release determined by matrix erosion. The rate constants of the release process of [AuCys]_n_ from PLA films during the first hour and after two days of incubation in buffer solution at 37 °C were measured. The tangent of the angle of slope of each line represents the effective rate constant of the additive release process, the values of which are presented in Table 2.

Thus, the data obtained in this work allow us to conclude that films containing a functional additive of gold–cysteine complex can release the additive in a controlled manner not only at pH 9, but also at pH 7, which demonstrates their potential for use as bioresorbable materials with biological activity in BNCT. In addition, the key parameters, such as the amount of released additive and its release rate, are quite close in both neutral and slightly alkaline medium ($k_{eff}=\frac{\Delta m}{\Delta t}\approx 1$ μg/h), which indicates the effectiveness of using the mechanism of crazing to create materials with controlled release.

It was also discovered that after 3 h of release of [AuCys]_n_ from the composite into pH 7 medium, the average particle size was found to be of the order of 30 nm (Figure 10), which is a good result for application in cancer therapy, as mentioned above. Under such conditions, without obtaining a nanocomposite, as shown in Figure 1, the particle size would be about 350 nm. Due to the formation of nanoparticles of the complex in the polylactide matrix, the is released into solution also in the form of nanoparticles, which are able to enlarge with time. However, during the first hours of release into a pH 7 medium, the particle size remains small, which is an excellent result for penetration into the cancer cell. At the same time, under body conditions, due to constant blood flow, the particles would not have time to aggregate over such a period and would remain quite small.

Ideally, the composite material should degrade by the end of therapy, so it is necessary to understand the kinetics of its degradation under conditions close to those of the organism. For this purpose, a sample of the nanocomposite was incubated in water at pH = 7 and 37 °C for 4 weeks. However, it is also important to understand that the system we obtained was a model system, and in the conditions of the organism, which would contain a sufficiently large number of biologically active substances, such as enzymes, degradation may occur at different rates. The determination of the number-average molecular weight (*M*_n_) was carried out before hydrolysis and then at the end of each week using the gel permeation chromatography (GPC) method. Figure 11 shows the dependence of *M*_n_ on soaking time in water. Three periods can be distinguished on the kinetic curve in accordance with the dependences that were previously described [35]:(1)Polylactide swelled in water for the first week, and there was practically no decrease in molecular weight during this period.(2)After the end of the first period, hydrolysis of PLA ester bonds occurred during the next 2 weeks, resulting in a 20% decrease in *M*_n_, with low molecular weight hydrolysis products remaining within the material.(3)At the third stage, a drop in the degradation rate was observed due to the shift of thermodynamic equilibrium towards high molecular weight compounds, which was in turn due to the increased concentration of low molecular weight compounds in the surrounding aqueous medium.

The kinetics of hydrolysis were characterized using the Lyu, Sparer and Unterecker model [35], and its parameters are given in Table 3. This model assumes that the degradation process of PLA is described by second-order reaction kinetics. The dependence of the inverse number of bonds in the polymer chain *N* as a function of time *t* is described by the following equation:$$\frac{1}{N\;\left(t\right)}=k_{2}\;\cdot \;C_{s}\;\cdot \;\left(t-t_{i}\right)$$
where $C_{s}$ is the water solubility in the polymer (constant), and $t_{i}$ is the effective induction time of volume erosion; if the bond breakage is much slower than the saturation process, then $t_{i}$ is the time for which the polymer is saturated with water (phase 1). 

This equation does not take into account the hydrolysis occurring during phase 1, and ceases to be applicable at the beginning of phase 3. At the same time, the initial value of molecular weight did not significantly affect the rate of PLA decomposition during phase 2, but it affected the duration of all erosion phases. 

Comparing the obtained data with the results of release, we can conclude that after 3 weeks, both [AuCys]_n_ release and PLA hydrolysis significantly slow down. In this case, possible products of polylactide hydrolysis, such as lactic acid derivatives, remained in solution, and an excess chemical potential appeared that shifted the thermodynamic equilibrium towards high molecular weight products. The conditions in a living organism would be somewhat different due to the washing out of low-molecular compounds in the implant area into the bloodstream, phase 3 would be absent from the PLA hydrolysis curve, and the degradation would occur more rapidly.

According to the results of the study by the thermogravimetric analysis (TGA) method in the samples that were incubated at pH 7 and 9, the [AuCys]_n_ content by the end of the experiment was slightly less than 6 wt%, while the initial content was 6.5 wt%. This fact is evidence that most of the filler remained inside the film PLA. However, as mentioned above, PLA hydrolysis in the body would have a different profile, so the additive would be released more efficiently in vivo. Such studies will be the subjects of our future work.

Thus, polylactide films containing complex gold salts obtained using structural and mechanical modification by the crazing method may be promising for use in medicine as bioresorbable materials with therapeutic agents for application in BNCT with controlled release time of functional additives, including the delayed effect.

[AuCys]_n_ showed dose-dependent cytotoxicity. Cell viability after 72 h of incubation with [AuCys]_n_ in a wide range of concentrations is shown in Figure 12. Human U251 glioblastoma cell line was more sensitive to the drug. A cytotoxic effect was observed at a concentration of 12.5 μg/mL, with the maximum safe concentration of gold being 6.25 μg/mL. The viability of SW-620 was not affected up to the concentration of 25 μg/mL, while the cytotoxic effect was noticed at a concentration of 50 μg/mL. Such significant differences in cell viability may be connected with cell line type.

The survival fraction of U251 and SW-620 after incubation with [AuCys]_n_ in concentration of 50 μg/ml is shown in Figure 13. These results confirm the results obtained in the methyl tetrazolium test (MTT). The drug significantly reduced the ability of both cell lines to form a colony. This effect was more prominent for U251, where cell survival was 51.2% compared to the untreated control, while the survival fraction of SW-620 was 73.2%, which was also significantly different from the control group. [AuCys]_n_ cytotoxicity is generally considered to be low, but the higher [AuCys]_n_ cytotoxicity in our case could be related to low culture medium concentrations that influenced cell nutrition and decreased proliferation when larger amounts of [AuCys]_n_ were added.

## 3. Materials and Methods

### 3.1. Materials

#### 3.1.1. PLA-Based Film Materials

In this work, we used 100 μm thick PLA films obtained from commercial granules of polymer grade 4032D (Nature Works LLC, Blair, Nebraska, USA) with the following molecular weight and physical characteristics: number-average molecular weight, *M*_w_ 95 kDa; dispersity, *Đ* 1.7; glass transition temperature, 60–63 °C; melting point, 167 °C. Amorphous PLA films were obtained by hot pressing at 190 °C and 150–170 kg·s·cm^−2^ pressure with rapid cooling with cold (~15 °C) water. Crystallization of the initial amorphous film was carried out in ethanol at 50 °C for 45 min.

#### 3.1.2. Obtaining Nanoporous PLA Films

Ethanol (95 wt% solution) was used as an adsorption-active medium. Porous film materials based on PLA were obtained by orientational stretching of the initial films in AAM to the tensile strain *ε* = 100% by the delocalized crazing mechanism [11] at a stretching rate of 25%/min. The calculation of the tensile strain was carried out according to the following equation: $\varepsilon =\frac{l_{f}-l_{o}}{l_{f}}\;\cdot \;100$, where $l_{f}$ is the final length, and $l_{o}$ is the initial length of the sample.

#### 3.1.3. [AuCys]_n_ Synthesis

[AuCys]_n_ in aqueous solution was prepared by reaction of HAuCl_4_ (chemically pure grade, Himmed, Moscow, Russia) with cysteine (chemically pure grade, Himmed, Moscow, Russia) in a molar ratio of 1:3. In brief, 100 mg [AuCys]_n_ was dissolved in 10 mL Milli-Q water and a solution of 107 mg cysteine in 10 mL Milli-Q water was added under stirring. The method we chose for obtaining the complex was in agreement with the previously described procedure [36].

An amount of 1M NaOH solution was used to achieve the desired pH value, as demonstrated in Figure 1.

#### 3.1.4. Obtaining PLA-[AuCys]_n_ Composites

Synthesis of [AuCys]_n_ was carried out in situ in open-porous PLA films by countercurrent diffusion method (the experimental scheme is shown in Figure 14), using 0.1 wt% aqueous solutions of NaAuCl_4_ (chemically pure grade, Himmed, Moscow, Russia) and cysteine (chemically pure grade, Himmed, Moscow, Russia) as initial reagents. The reaction was carried out in a dark room at room temperature for 1 day. Polymer samples containing [AuCys]_n_ were thoroughly washed in deionized water and dried to constant weight under standard conditions.

### 3.2. Research Methods

#### 3.2.1. UV-Vis Spectroscopy

The process of release of [AuCys]_n_ from PLA matrices into a buffer medium was studied by UV-Vis spectroscopy on a Thermo Heλios α device; the scanning rate was 100 nm/min. The process was carried out at 37 °C in 0.2 M Na_2_HPO_4_ solution (chemically pure grade, Himmed, Moscow, Russia) with pH = 9 and in deionized water with pH = 7. For this purpose, samples of composite films weighing about 20 mg were placed in 20 mL vials, which were filled with Na_2_HPO_4_ solution or water and tightly closed with lids. The vials were heated under constant stirring on a heating plate IKA (Staufen, Germany) with a contact thermometer and using a water bath. Over two weeks at regular intervals, 0.15 mL amounts of the solutions were sampled from the vials and placed in quartz cuvettes with Teflon lids, with the addition of either 1.35 mL of 0.2 M Na_2_HPO_4_ solution with pH 9, or 1.35 mL of water with pH 7.

#### 3.2.2. Thermogravimetric Analysis

[AuCys]_n_ content was determined by TGA on a Mettler TA4000 instrument (Mettler Instruments, Greifensee, Switzerland). Samples with a diameter of 6.5 mm were pre-cut from the composites and placed in a ceramic crucible. Then they were heated in the temperature range from 25 to 700 °C at a rate of 20 °C/min in air.

#### 3.2.3. Differential Scanning Calorimetry

The thermal properties of the polymer samples were studied via DSC with a Metler TA4000 instrument (DSC–20 cell) (Metler Toledo, Greifensee, Switzerland) in the temperature range 25–200 °C at a heating rate of 10 °C min^−1^ in a nitrogen atmosphere.

The degree of crystallinity (α) of the samples was determined using the next equation:$$\alpha =\frac{\Delta H_{\mathrm{melt}}-\Delta H_{\mathrm{cryst}}}{\Delta H_{\mathrm{melt}}\left(100\%\right)}$$
where ∆H_melt_ is the enthalpy of melting, ∆H_cryst_ is the enthalpy of crystallization and ∆H_melt_(100%) is the melting enthalpy of a polymer with a degree of crystallinity of 100% (for PLA ∆H_melt_(100%) = 93 J∙g^−1^ [37]).

#### 3.2.4. Electron Microscopy

The morphology of samples was studied by SEM using a JEOL JSM-6380LA (Tokyo, Japan) microscope at an operating voltage of 20 kV. Preliminarily, polymer cleavages were prepared by brittle fracture in liquid nitrogen, and were then attached to a microscopic holder with carbon tape and sputtered with a 25 nm thick gold layer on an Eiko IB-3 Ion Coater (Tokyo, Japan).

The samples were studied by transmission electron microscopy, and electron scattering patterns were obtained using the microscope “LEO 912 AB OMEGA” (Zeiss, Germany). For this purpose, ultrathin sections (about 100 nm thickness) of composite films were obtained by ultramicrotomy with a diamond knife and placed on a copper grid coated with a collodion substrate. The determination of interplanar distances from the electron diffraction pattern was carried out using a standard gold sample.

#### 3.2.5. Atomic Force Microscopy 

The surface morphology was studied with a Multimode microscope with a Nanoscope V controller (Bruker, New York, NY, USA) using silicon cantilevers with a high aspect ratio and a resonant frequency of 70 kHz with a Q-factor of more than 300 (TipsNano, Tallinn, Estonia). The studies were carried out at room temperature under the tapping mode. Image processing was performed using femtoscan.2.4.26 (FemtoScan Online software, Advanced Technologies Centre, Moscow, Russia) [38].

#### 3.2.6. Analysis of Molecular Weight Characteristics of PLA

The kinetics of hydrolytic degradation of PLA were studied by measuring the change in the number-average molecular weight *M*_n_ of the polymer with time. For this, the composite film was kept in deionized water at pH = 7 and 37 °C for 4 weeks. *M*_n_ of PLA were determined by GPC on a 1260 Infinity II GPC/SEC Multidetector System (Agilent, Santa Clara, CA, USA) chromatograph equipped with refractive and viscometric detectors, a light scattering detector, and two PLgel 5 μm MIXED B columns. Tetrahydrofuran was used as solvent. Molecular weight was determined according to the calibration curve using narrowly dispersed PMMA standards (M = 5 × 10^2^–1 × 10^7^). 

#### 3.2.7. Dynamic Light Scattering (DLS)

Dynamic light scattering experiments were carried out using a (Litesizer 500, Anton Paar, Turin, Italy). Data acquisition parameters were set as follows: time of analysis, 3 min (average of 3 measurements of 1 min); scattering angle, θ = 90°; temperature, T = 25 °C.

#### 3.2.8. Statistical Analysis

The value of any parameter was measured 2–3 times for all types of studies in the work. Statistical processing of the data was carried out according to analysis of variance using a test of least significance and comparing the difference between the means of different groups. The significance test was determined at *p* < 0.05. The data values and error bars on the graphs are presented as the mean ± standard deviation.

### 3.3. Cytotoxicity Assay

#### 3.3.1. Cell Lines

In vitro experiments were carried out in U251 (human glioblastoma) and SW-620 (human colorectal adenocarcinoma) cell lines, which were purchased from the Center for Genetic Resources of Laboratory Animals of the Institute of Cytology and Genetics of the Russian Academy of Sciences (Novosibirsk, Russia). Cells were cultivated in DMEM/F12 (1:1) (Biolot, Moscow, Russia) supplemented with 10% fetal bovine serum (HiMedia Laboratories, Mumbai, India) and gentamicin 50 mg/mL (Dalkhimpharm, Moscow, Russia) at 37 °C and 5% CO_2_.

#### 3.3.2. MTT Test

For the cytotoxicity studies, an aqueous solution of pH 7, into which the [AuCys]_n_ complex from the composite was released for one month, was freeze-dried. The resulting powder was diluted to the concentrations shown in Figure 12 and Figure 13. The assessment of cell viability after incubation with nanoparticles for 72 h was performed using an MTT-test. Cells were seeded in 96-well plates at a density of 2 × 10^4^ cells per well. After 24 h, [AuCys]_n_ was added in concentrations of 1.5–100 μg/mL and incubated for 72 h. Control wells were incubated without the drug. Then, the medium in all wells was replaced with 100 µL of medium and 20 µL of a solution of 3-(4,5-dimethylthiazol-2-yl)-2,5-diphenyltetrazolium bromide) (MTT). The plates were incubated under standard conditions for 4 h, and the medium with MTT was replaced with dimethyl sulfoxide. The results were analyzed spectrophotometrically with a Multiskan SkyHigh (Thermo, Waltham, MA, USA) at a wavelength of 595 nm. Data are presented as a percentage of the viability of each control.

#### 3.3.3. Colony Forming Assay

The cells were cultivated in culture flasks. In the logarithmic phase of growth, the [AuCys]_n_, obtained as described in Section 3.3.2, was added to the final concentrations in 50 μg/mL of culture medium and incubated for another 24 h. Control samples were incubated without the drug. Then the cells were removed and seeded in 6-well plates at density of 200 cells per well. The plates were cultivated under standard conditions over the time period to form the colonies. Eight days after drug administration, the colonies were fixed with 10% formalin (Panreac AppliChem, Darmstadt, Germany), stained with crystal violet 0.5% water solution (Panreac AppliChem, Darmstadt, Germany) and dried. Counting of colonies (≥50 cells) was performed using a light inverted microscope Zeiss Primo Vert (Zeiss, Germany). The results of the experimental groups were compared with the data of the control groups.

## 4. Conclusions

The optimization of conditions for the synthesis of a cysteine–gold complex compound was carried out. The method of controlling the particle size and stability of gold complex compounds by changing the pH was shown for the first time. With increasing pH, the stability of the solution increased from several minutes to weeks. Moreover, there was a linear decrease in particle size from hundreds to tens of nanometers due to the formation of a soluble form of the [Au(SR)_2_]^−^ type complex in alkaline medium.The content, size and character of the distribution of gold complex compounds synthesized in the volume of nanoporous PLA matrices was determined. The ability to control the formation of gold complex salt particles within the pore size of the polymer was demonstrated for the first time. It was found that the synthesis of gold complex salts in the macroscopic volume led to the formation of particles of about 350 nm in solution without polymer stabilization, while the particles were 5–20 nm in size when directly synthesized in the pores of the polymer matrix obtained by the crazing mechanism. The sizes of the formed gold complex compound particles were retained after their release from the polymer pores. The results obtained are important for application in BNCT dosimetry, since particles smaller than 100 nm are desirable for this task.The possibility of prolonged release of gold complex compound from the obtained composite was shown. The release of the filler from the polymer matrix volume had a delayed start—this process began only after several hours and was characterized by an effective rate constant of 1 µg/h. At the same time, in vitro studies showed that the concentration of 6.25 µg/mL was reliably safe and did not reduce the survival of U251 and SW-620 cells. Simultaneously with the release of the filler, decomposition of the PLA matrix occurred.

## Figures and Tables

**Figure 1 ijms-24-16492-f001:**
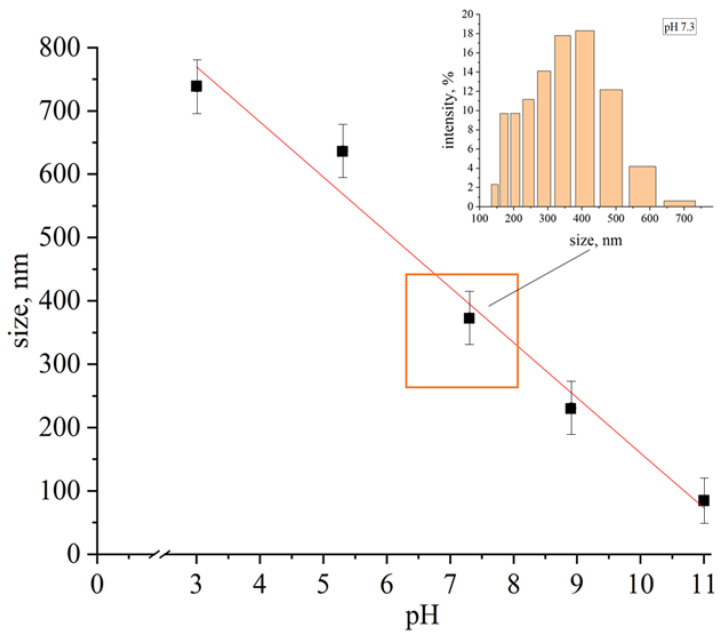
Dependence of the average size of the gold–cysteine [AuCys]_n_ particles versus pH.

**Figure 2 ijms-24-16492-f002:**
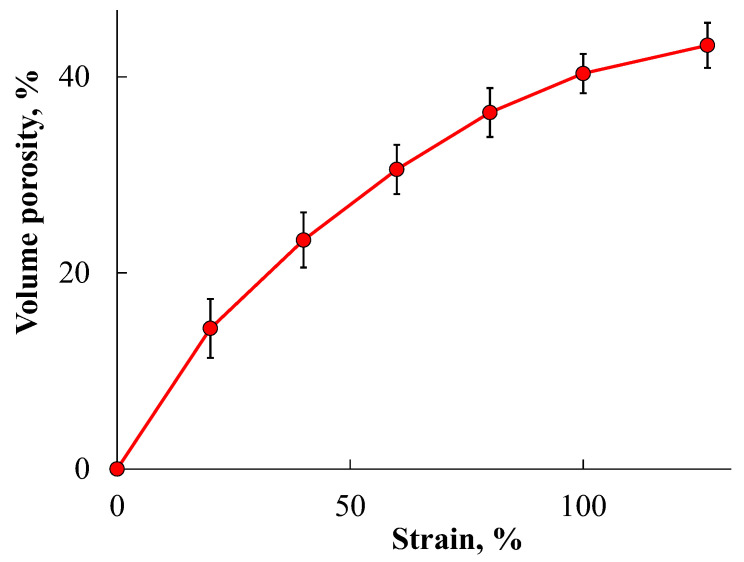
Dependence of crystalline PLA volume porosity vs. the tensile strain during stretching in ethanol.

**Figure 3 ijms-24-16492-f003:**
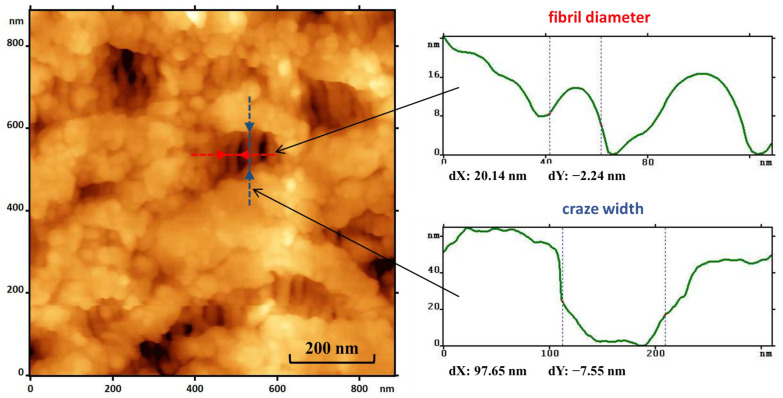
AFM image of a PLA film deformed by 150% in ethanol. The insets show the height distributions along the section lines indicated in the image.

**Figure 4 ijms-24-16492-f004:**
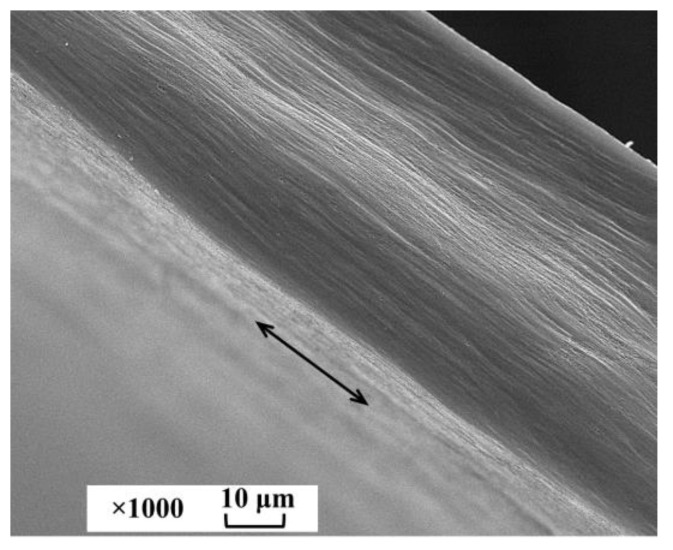
SEM micrograph of a brittle cleavage of a porous PLA film with a tensile strain of 100%. The arrow indicates the direction of stretching.

**Figure 5 ijms-24-16492-f005:**
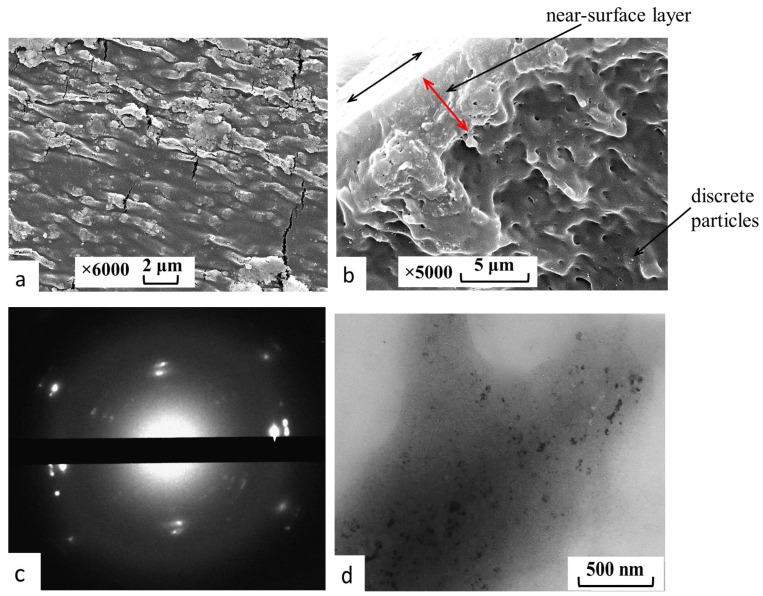
(**a**,**b**) SEM and (**d**) TEM micrographs of (**a**) surface, (**b**) brittle cleavage and (**d**) thin section of PLA films containing [AuCys]_n_. (**c**) Electron diffraction pattern obtained from sample (**d**). The two-way black arrow shows the thickness of the layer.

**Figure 6 ijms-24-16492-f006:**
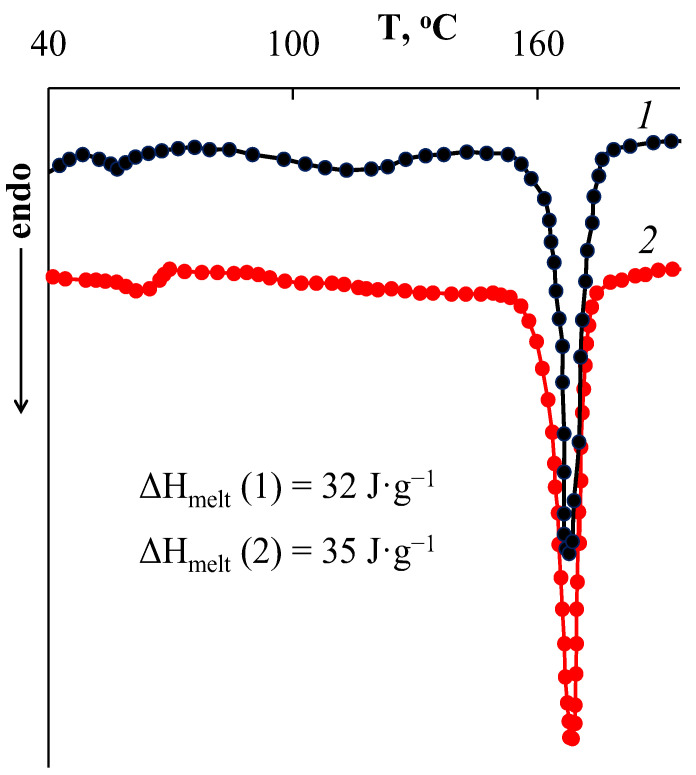
DSC curves of the initial PLA matrix (1) and PLA containing [AuCys]_n_ (2).

**Figure 7 ijms-24-16492-f007:**
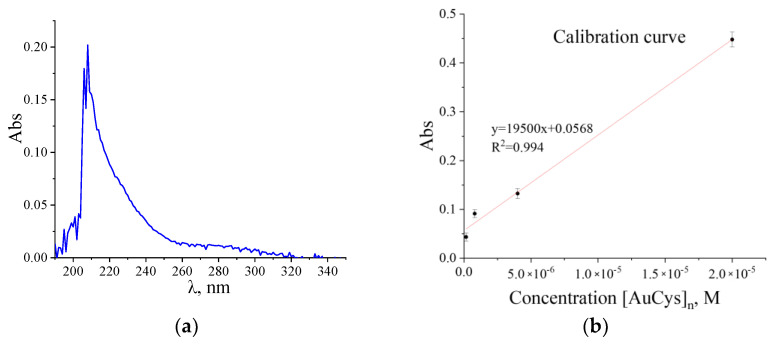
(**a**) Characteristic UV spectrum of the [AuCys]_n_ complex; (**b**) calibration plot for the determination of [AuCys]_n_ content in PLA samples.

**Figure 8 ijms-24-16492-f008:**
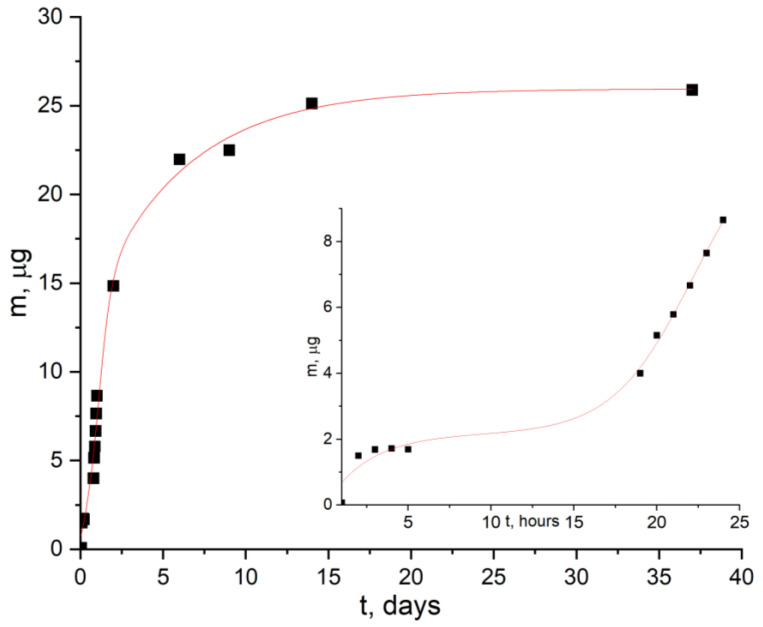
Kinetic curve of [AuCys]_n_ release from PLA-based film composite at pH 7 and its approximation by the Gallagher–Corrigan model. The inset shows the initial section of the curve.

**Figure 9 ijms-24-16492-f009:**
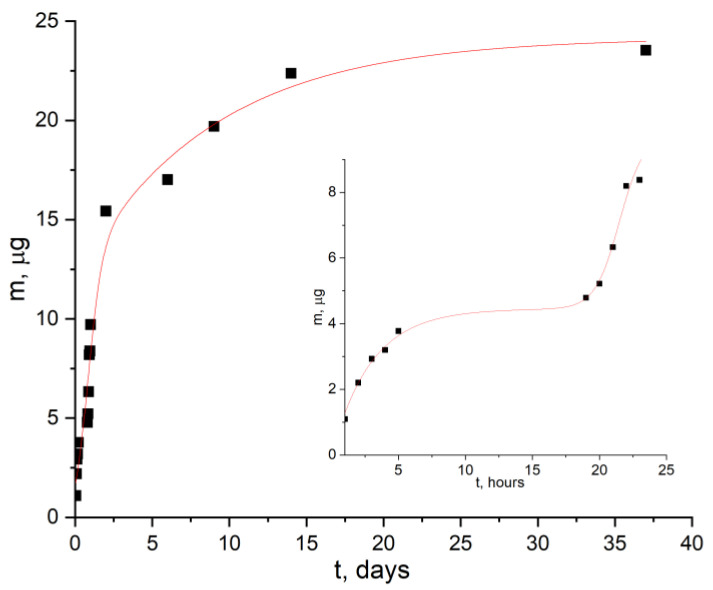
Kinetic curve of [AuCys]_n_ release from PLA-based film composite at pH 9 and its approximation by the Gallagher–Corrigan model. The inset shows the initial section of the curve.

**Figure 10 ijms-24-16492-f010:**
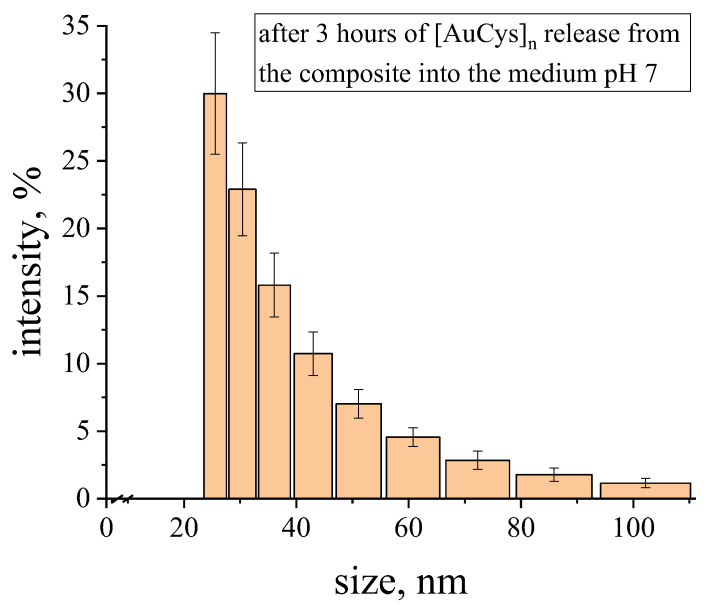
Particle size distribution of the [AuCys]_n_ complex, which was released from PLA-based film composite at pH 7 for 3 h.

**Figure 11 ijms-24-16492-f011:**
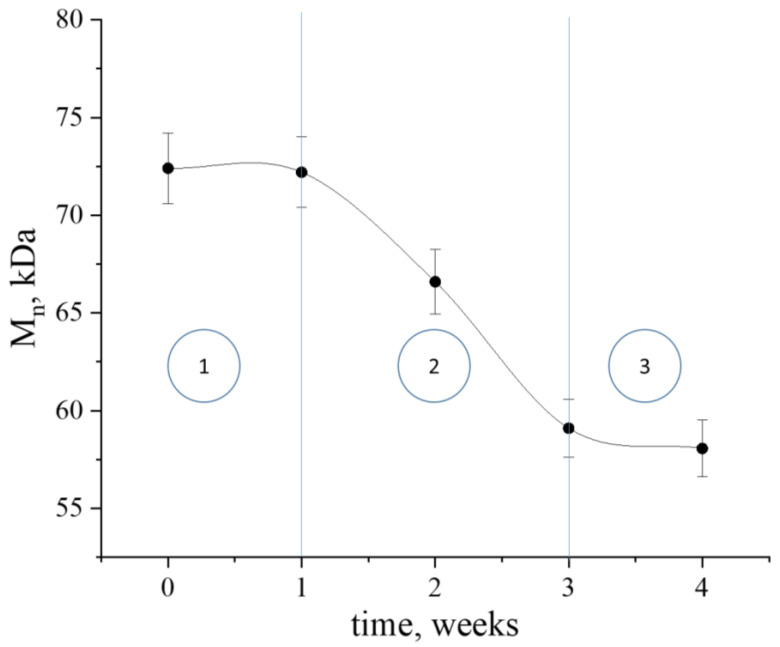
Number-average molecular weight (Mn) of PLA as a function of time.

**Figure 12 ijms-24-16492-f012:**
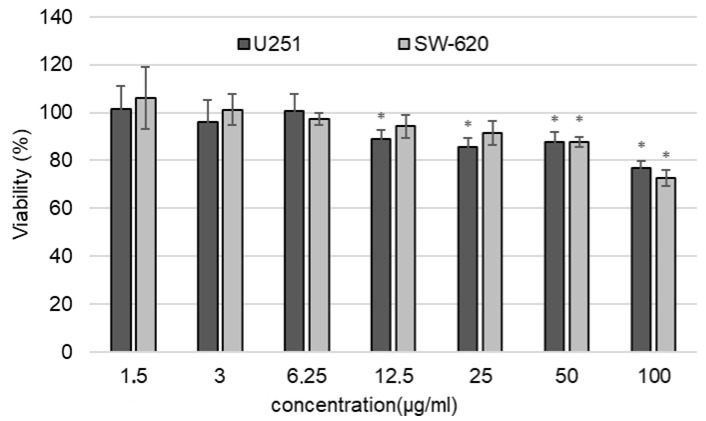
Evaluation of the dose-dependent cytotoxicity of the [AuCys]_n_ after 72 h in U251 and SW-620 cell lines. * *p* < 0.05—different from the control group (Mann–Whitney U test).

**Figure 13 ijms-24-16492-f013:**
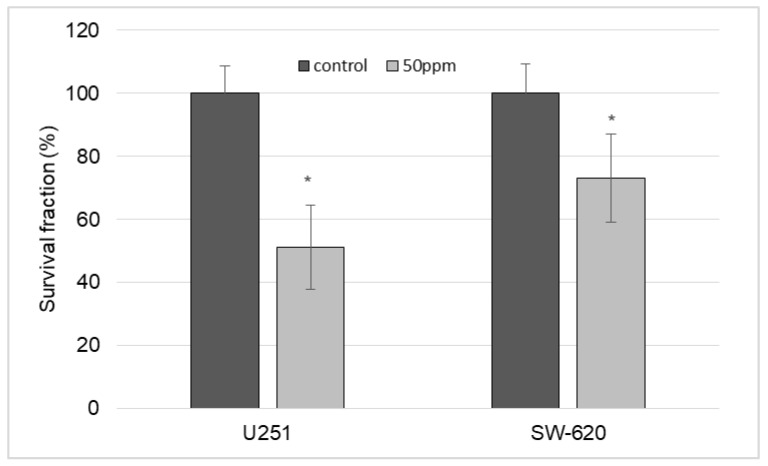
Determination of the effect of the [AuCys]_n_ at a concentration of 50 µg/mL on the colony-forming capacity of U251 and SW-620 cell lines. * *p* < 0.05—different from the control group (Mann–Whitney U test).

**Figure 14 ijms-24-16492-f014:**
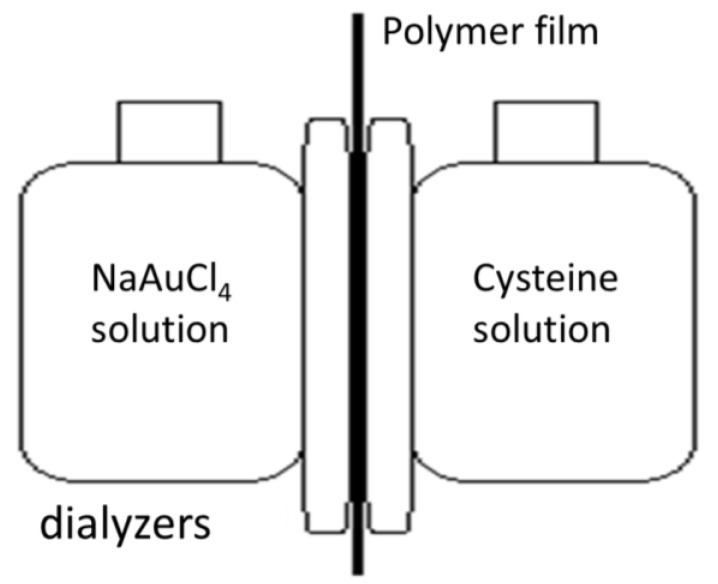
Scheme of [AuCys]_n_ synthesis inside the polymer film by the countercurrent diffusion method.

**Table 1 ijms-24-16492-t001:** Gallagher and Corrigan model equations.

Equation	Equation Parameters
$f_{t}=f_{t,max}\left(1-e^{-k_{1}t}\right)+\left(f_{t,max}-f_{B}\right)(\frac{e^{k_{2}t-k_{2}t_{2,max}}}{1+e^{k_{2}t-k_{2}t_{2,max}}}$)	$f_{b}$—the amount of released additive in the first stage (explosive effect) $k_{1}$—first-order rate constant at the first stage
	$k_{2}$—second stage rate constant
	$t_{2,max}$—time of maximum release rate at the second stage data

**Table 2 ijms-24-16492-t002:** Gallagher–Corrigan model parameters determined for the release process of [AuCys]_n_ from the composite under pH 7 and pH 9 conditions at 37 °C.

pH	t_2_, Max, h	f_t,max_, μg	f_b_, μg	k_1_, h^−1^	k_2_, h^−1^	R^2^
7	27 ± 2	25.9	2.0	0.007 ± 0.001	0.11 ± 0.02	0.99
9	21 ± 4	23.5	4.5	0.005 ± 0.002	0.10 ± 0.02	0.98

**Table 3 ijms-24-16492-t003:** Parameters of the Liu, Sparer and Unterecker model for the degradation kinetics of PLA in a composite containing [AuCys]_n_ in deionized water at pH 7 and 37 °C.

Model	$C_{s}$, mol/L	$t_{i}$, ч	R^2^	$k_{2}$, L/(mol·h)
$\frac{1}{N\;\left(t\right)}=k_{2}\;\cdot \;C_{s}\;\cdot \;(t-t_{i}$)	3.42	168	0.98	2 · 10^−8^ ± 1 · 10^−8^

## Data Availability

The data presented in this study are available on request from the corresponding author.

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
