# Peer review of "Nanocomposite Materials Based on Polylactide and Gold Complex Compounds for Absorbed Dose Diagnostics in BNCT"

_ijms, 2023, doi:10.3390/ijms242216492_

Round 1
Reviewer 1 Report
Comments and Suggestions for Authors
After careful review, I realized that authors need to make major revisions prior to acceptance. The following are the points that need to be addressed
i) The primary goal of the study is to develop a diagnosis tool but, in the manuscript, there is information on the application of new materials as diagnostic tools.
ii) The author should check the MTT assay against normal cells as well to confirm the cell-specific cytotoxicity study.
iii) The authors need to check [AuCys]n release study at acidic pH (~6.5 pH) as acidity is the characteristic feature of cancer cells.
iv) The [AuCys]n particle size (300 nm) is higher in physiological pH, then, how the author prepared a lower size of nano complex with a polymer scaffold.
Author Response
Dear Editor,
Thank you very much for your kind consideration and suggestions for improving the Manuscript. We have done our best to improve it according to your and reviewer's comments.
i) The primary goal of the study is to develop a diagnosis tool but, in the manuscript, there is information on the application of new materials as diagnostic tools.
This study aims to develop a material that can be used in the BNCT diagnosis. BNCT is a rapidly developing field that has been in use for less than 5 years and currently no approach has been developed to measure absorbed dose in BNCT, but there are studies suggesting that gold may be a potential agent for this task. We have made changes to the text that clarify our position.
ii) The author should check the MTT assay against normal cells as well to confirm the cell-specific cytotoxicity study.
Yes, we agree with this remark. Since this was a pilot study, we only used 2 tumor cell lines to determine how cytotoxic the drug is, however we are planning to continue working with [AuCys]n and include more cell lines in the study, in particular normal fibroblasts.
iii) The authors need to check [AuCys]n release study at acidic pH (~6.5 pH) as acidity is the characteristic feature of cancer cells.
Before measuring the release kinetics, we assumed that pH would have a significant effect on the release rate. However, it turned out that at both pH 7 and pH 9 the results were quite similar, which may indicate a weak dependence of the release kinetics on pH. We chose pH 7.0 as something intermediate between 6.5, characteristic of cancer cells and 7.4 corresponding to blood pH for better representativeness.
iv) The [AuCys]n particle size (300 nm) is higher in physiological pH, then, how the author prepared a lower size of nano complex with a polymer scaffold.
The particle size in the nanocomposite where PLA is used as matrix corresponds to the pore size in the matrix and it appears to be an order of magnitude smaller than for synthesis in solution. In the paper we discuss that «The most appropriate way to produce nanocomposites based on PLA and complex is the synthesis of [Au(I)-SR]n in a previously prepared nanoporous film, since in this case the particle size of the complex is limited by the pore size, and the process itself can be carried out under room temperature conditions.»
Next in the paper we show, that the pore size is of the order of 15-20 nm. As a result, the particle size of [AuCys]n also turns out to be about 20 nm. Pore size limits particle growth [AuCys]n.
Many thanks again for your hard work.
On behalf of all authors,
Potseleev Vladislav
Reviewer 2 Report
Comments and Suggestions for Authors
1. Did the authors use any reducing agent during the [AuCys]n synthesis via the in-site method?
2. Please update Section 2.3.1 with an appropriate title
3. Section 2 focuses primarily on materials and methods rather than experiment results. Suggest moving lines 171-173.
4. Lines 174 to 205 seem to provide background information. Suggest relocating them to Section 1.
5. Could the authors clarify the intended application of this composite complex? The title suggests diagnostic use, while the context mentions its potential in BNСT for delayed recurrence therapy.
6. Did the authors test the synthesized composite on normal/healthy cells for cellular safety assessment?
7. In Figure 13, it would be helpful to include a legend describing the bars.
Comments on the Quality of English LanguageThe sentences within this manuscript are highly declarative and do not offer proper transitions. For instance, most of the abstract was primarily descriptive and regarding what was done, rather than describing the rationale for the experiments, the findings, and the implications of the findings. While this writing is technically not incorrect, it negatively impacts the quality of the writing and does not adequately engage the readers.
Author Response
Dear Editor,
Thank you very much for your kind consideration and suggestions for improving the Manuscript.
We have done our best to improve it according to your and reviewer's comments.
- Did the authors use any reducing agent during the [AuCys]n synthesis via the in-site method?
The article states «[AuCys]n formed by the interaction of aqueous solutions of HAuCl4 and L-cysteine». In this case, L-cysteine is used as both a reducing agent and a complexing agent.
We have added Section 3.1.3. which describes in more detail the method for producing [AuCys]n in solution
- Please update Section 2.3.1 with an appropriate title
Yes, we agree with this remark. We changed the title.
- Section 2 focuses primarily on materials and methods rather than experiment results. Suggest moving lines 171-173.
We agree. We removed this text
- Lines 174 to 205 seem to provide background information. Suggest relocating them to Section 1.
We moved the text to Section 1.
- Could the authors clarify the intended application of this composite complex? The title suggests diagnostic use, while the context mentions its potential in BNСT for delayed recurrence therapy.
We propose to apply this composite material for the therapy of delayed relapses in the BNCT technique, where gold will act as an agent for dosimetry. The next stage of our work will include studies aimed at synthesising gold/boron single molecule or nanocomposite containing gold/boron. The title of the work reflects the direction of research and the purpose in the application of the synthesised materials.
- Did the authors test the synthesized composite on normal/healthy cells for cellular safety assessment?
Since this study was a pilot study, we only used 2 tumor cell lines to determine the cytotoxicity of the nanocomposite, and did not investigate the effect on normal cell lines. Further we are planning to continue working with [AuCys]n and include more cell lines in the study, in particular normal fibroblasts.
- In Figure 13, it would be helpful to include a legend describing the bars.
Thanks for the note, we fixed it.
We've also added more transitions between some sections and sentences.
Many thanks again for your hard work.
On behalf of all authors,
Potseleev Vladislav
Reviewer 3 Report
Comments and Suggestions for Authors
The findings presented in your manuscript do not significantly contribute to the existing body of knowledge in the field. The research does not introduce new concepts, methodologies, or insights that would advance the field or have a meaningful impact. The study appears to replicate or reaffirm well-established findings without providing any novel or substantial contributions.
The structure and clarity of the manuscript need significant improvement. The manuscript lacks a logical flow, making it challenging to follow the research design, methods, and results. The writing style is unclear and convoluted, impeding the effective communication of the research.
Abstract
1. The abstract jumps straight into discussing the synthesis of gold compounds without providing sufficient background information or explaining the significance of the research.
2. The abstract mentions the synthesis of complex compounds of gold with cysteine but does not provide details about the specific compounds synthesized or their intended applications.
3. The abstract includes phrases such as “this process begins only after a few days and is characterized by an effective rate constant of 1 19 μg/h”. without providing clear explanations or definitions, making it difficult for readers to fully understand the findings.
4. The abstract states that the release kinetics of complex gold salts from the polymer matrix were investigated, but it later refers to the release of [AuCys]n from the polymeric matrix, which creates confusion about the exact compound being studied.
Introduction
1. The introduction jumps from discussing boron neutron capture therapy (BNCT) to the problems associated with gold nanoparticles (NPs) in BNCT without a smooth transition or clear connection between the topics.
2. The introduction states that the synthesis of safe drugs with high neutron absorption is the main scientific problem that needs to be solved, but it does not clearly articulate the specific research problem being addressed in this study.
Results and Discussion
1. The section lacks specific details about the experimental procedures, such as the encapsulation methods used, characterization techniques employed, and the parameters measured. Providing more specific information would enhance the reproducibility and transparency of the study.
2. The paper primarily relies on SEM, TEM, AFM, and XRD techniques for characterizing the nanocomposites. While these techniques provide valuable information, they are insufficient for a comprehensive characterization of the materials. Additional techniques, such as FTIR and thermal analysis, would provide a more thorough understanding of the composition, structure, and thermal properties of the nanocomposites.
3. The section does not provide a comprehensive discussion of the implications and significance of the obtained results. It does not address how the findings contribute to the existing knowledge or how they relate to the broader field of polymer nanocomposites. A more in-depth discussion would help readers understand the importance of the research and its potential applications.
4. The section does not include any statistical analysis method of the data presented.
5. The section does not provide a comprehensive comparison with relevant literature. It fails to discuss how the obtained results align with or differ from previous studies in the field. Including such a comparison would enhance the scientific discourse and help readers understand the novelty and significance of the research.
Comments on the Quality of English Language
Pay attention to grammatical errors, including issues with verb agreement, subject-verb agreement.
Author Response
Dear Editor,
Thank you very much for your kind consideration and suggestions for improving the Manuscript.
We have done our best to improve it according to your and reviewer's comments.
- The abstract jumps straight into discussing the synthesis of gold compounds without providing sufficient background information or explaining the significance of the research.
We agree with the remark. We have amended the abstract to clarify our position.
- The abstract mentions the synthesis of complex compounds of gold with cysteine but does not provide details about the specific compounds synthesized or their intended applications.
We have modified this part of the Abstract slightly and added the part of obtaining [AuCys]n in solution to the Materials and Methods section
- The abstract includes phrases such as “this process begins only after a few days and is characterized by an effective rate constant of 1 19 μg/h”. without providing clear explanations or definitions, making it difficult for readers to fully understand the findings.
We have slightly changed and clarified this point in the text.
- The abstract states that the release kinetics of complex gold salts from the polymer matrix were investigated, but it later refers to the release of [AuCys]n from the polymeric matrix, which creates confusion about the exact compound being studied.
We agree with the remark, we have corrected the ambiguities.
Introduction
- The introduction jumps from discussing boron neutron capture therapy (BNCT) to the problems associated with gold nanoparticles (NPs) in BNCT without a smooth transition or clear connection between the topics.
We agree with the comment, and have added a smoother transition and explanation of the significance of the complex [AuCys]n for BNСT applications
- The introduction states that the synthesis of safe drugs with high neutron absorption is the main scientific problem that needs to be solved, but it does not clearly articulate the specific research problem being addressed in this study.
We have made significant changes to Section 1 to clarify the scientific problem
Results and Discussion
- The section lacks specific details about the experimental procedures, such as the encapsulation methods used, characterization techniques employed, and the parameters measured. Providing more specific information would enhance the reproducibility and transparency of the study.
Sections 3.1.2 and 3.1.4 describe the porous matrix preparation and encapsulation in detail. We have also added some clarifying points in the Results and Discussion section. Previously, a similar technique for the synthesis of nanoscale particles in the volume of porous polymer matrices obtained by the crazing mechanism was described in detail in ref. 47. Stabilisation and encapsulation of the formed particles occurs as a result of their adsorption on the surface of the nanoscale pore and fibril system.
- The paper primarily relies on SEM, TEM, AFM, and XRD techniques for characterizing the nanocomposites. While these techniques provide valuable information, they are insufficient for a comprehensive characterization of the materials. Additional techniques, such as FTIR and thermal analysis, would provide a more thorough understanding of the composition, structure, and thermal properties of the nanocomposites.
Yes, we agree with this remark. Added information about DSC in sections 2 and 3. We obtain composites with a thickness of about 100 microns. Thin films are the best option for FTIR. In our case, we can only look at the FTIR surfaces of the samples, which we believe is not quite representative.
- The section does not provide a comprehensive discussion of the implications and significance of the obtained results. It does not address how the findings contribute to the existing knowledge or how they relate to the broader field of polymer nanocomposites. A more in-depth discussion would help readers understand the importance of the research and its potential applications.
We have added in section 2 some comparisons with other studies in the field of nanocomposite materials production and the importance of our study.
- The section does not include any statistical analysis method of the data presented.
We have corrected the Materials and Methods and modified the graphs accordingly. The following part was added to the Materials and Methods.
- The section does not provide a comprehensive comparison with relevant literature. It fails to discuss how the obtained results align with or differ from previous studies in the field. Including such a comparison would enhance the scientific discourse and help readers understand the novelty and significance of the research
We agree with this remark. We have modified the text of the article and added comparisons with previous studies.
Many thanks again for your hard work.
On behalf of all authors,
Potseleev Vladislav
Round 2
Reviewer 1 Report
Comments and Suggestions for Authors
I am completely persuaded by the revisions made by the authors. The current version of the manuscript should be approved for publication.
Reviewer 2 Report
Comments and Suggestions for Authors
Thanks for addressing all the questions.
Comments on the Quality of English LanguageThe manuscript was written clearly.